# Interventions for vulnerable pregnant women: Factors influencing culturally appropriate implementation according to health professionals: A qualitative study

Esther I. Feijen-de Jong [1,2,3] *, J. Catja Warmelink [1,2,3], Relinde A. van der Stouwe [1,2,3], Maria Dalmaijer [1,2,3], Danielle E. M. C. Jansen [1,2]

1 Midwifery Science AVAG Section, Department of General Practice & Elderly Care Medicine, University of Groningen, University Medical Center Groningen, Groningen, The Netherlands, 2 Department of Midwifery Science, Amsterdam Public Health Research Institute, VU University Medical Center, Amsterdam, The Netherlands, 3 Midwifery Academy Amsterdam/Groningen, Groningen, The Netherlands

* e.i.feijen-de.jong@umcg.nl

**Data Availability Statement:** We are not willing to share the qualitative datasets (the interview transcripts) in the main paper or additional supporting files in order to protect participants'

## Abstract

### Background

Proper implementation of interventions by health professionals has a critical effect on their effectiveness and the quality of care provided, especially in the case of vulnerable pregnant women. It is important, therefore, to assess the implementation of interventions in care settings to serve as input to improve implementation.

### Objective

The aim of this study is to identify factors that influence the implementation of interventions for vulnerable pregnant women in the North of the Netherlands from the perspective of health professionals. In this region, an intergenerational transfer of poverty is apparent, leading to many health problems and the transfer of unhealthy lifestyles and the associated diseases to subsequent generations.

### Methods

We used a qualitative research design. Semi-structured interviews with 39 health professionals were conducted between February 2019 and April 2020. To analyse the findings, the MIDI (Measurement Instrument for Determinants of Innovations) was used, an instrument designed to identify what determinants influence the actual use of a new or existing innovation.

### Results

We found two themes that influence the implementation of interventions: 1. The attitude of health professionals towards vulnerable pregnant women: stereotyped remarks and words

confidentiality. This is in alignment with the requirements of the METC Groningen in which anonymity of participants must be guaranteed. Although we did remove personal identifiers from the interview transcripts (e.g. names and addresses), the transcripts are likely to contain references to the contextual identifiers in individual stories and make individuals identifiable. We can provide access to the transcripts and audit trail on request and subject to certain conditions. If you want to access data you can contact a researcher of the unit Midwifery Science via the general email address of the department of General Practice & Elderly Medicine: huisartsgeneeskunde@umcg.nl.

**Funding:** The work was supported by ZonMW | The Netherlands Organisation for Health Research and Development (project/grant number 50-54300-98-261). The grant was awarded to DJ. No authors received salary or other funding from commercial companies. The URL to the sponsor's website: https://www.zonmw.nl/en/ The funders had no role in study design, data collection and analysis, decision to publish, or preparation of the manuscript.

**Competing interests:** The authors have declared that no competing interests exist.

expressing the homogenization of vulnerable pregnant women. 2. A theme related to the MIDI determinants, under which we added six determinants.

## Conclusion

Our research showed that many factors influence the implementation of interventions for vulnerable pregnant women, making the optimal implementation of interventions very complex. We highlight the need to challenge stereotypical views and attitudes towards specific groups in order to provide relation-centred care, which is extremely important to provide culturally appropriate care. Health professionals need to reflect on their own significant influence on access to and the use of care by vulnerable groups. They hold the key to creating partnerships with women to obtain the best health for mothers and their babies.

## Introduction

Many interventions are developed in maternity care in order to improve perinatal and maternal health [1]. Some of them are specifically designed for vulnerable groups, which means that they take into account the preferences of vulnerable women and the cultures of their communities. Proper implementation of these interventions, in particular for vulnerable groups, has a critical effect on their effectiveness and the quality of care [2]. Implementation can be defined as 'the process of putting to use or integrating interventions within a setting' [3]. Within this implementation process, many influencing factors (i.e. factors related to the innovation (e.g. complexity or guidelines for using the innovation), the user (e.g.motivation, knowledge), the organization (e.g.resources), and the socio-political context (e.g.legislation)) [4] can consequently interfere with a possible reducing effect of the intervention [5]. In addition, especially the role of the professionals providing care might enhance the uptake of interventions [6]. It is important, therefore, to assess the implementation of interventions in actual care settings from the perspectives of health care providers [7].

The existing research focuses mainly on the development and validation of maternity care interventions and their outcomes [8–10] and the factors that influence the implementation of specific interventions in practice. However, research by Jones, Lattof and Coast [11] shows that a sound body of evidence regarding the factors that influence the implementation of culturally appropriate maternity care interventions, including stakeholders' perspectives, is lacking. They state that research has to move beyond recommendations that 'cultural factors should be taken into account'. Without appropriate cultural knowledge, it is difficult to match the needs, values and resources of healthcare providers and women [7] in specific contexts, which is vital to the use of interventions by healthcare professionals. Assessing implementation is therefore an absolute necessity in order to gain better outcomes for healthcare users living in vulnerable situations [5].

The aim of this study is to provide an overview of the factors that influence the implementation of interventions for vulnerable pregnant women from the perspective of health professionals. Our research question is: What are the factors influencing the implementation of interventions for vulnerable pregnant women from the perspective of health professionals?

## Study setting

In the North of the Netherlands, there is an intergenerational transfer of poverty in many cases [12], leading to many health problems and the passing on of unhealthy lifestyles and associated diseases to subsequent generations. This is reflected in the figures: women of reproductive age in the North are more likely to have adverse lifestyle characteristics than women in other parts of the Netherlands: more of them smoke, they have higher body weight and are more likely to drink alcohol compared with the national average [13]. The maternal social disadvantage is associated with poor health status in pregnancy, which in turn adversely affects birth outcomes [14]. Unlike in the urban, western part of the Netherlands, women living in poverty in the North are mostly of Dutch origin and live in relatively sparsely populated rural areas [15], which requires a specific approach towards the implementation of interventions that is sensitive to the characteristics of this group.

> Within the Netherlands, midwives are the lead medical professionals for providing care to women with 'normal' or uncomplicated pregnancies. They are independent practitioners (just like General Practitioners) who can work independently in a private midwifery practice or as part of a group. Almost 85% of all women start prenatal care with these midwives in the community [16].
>
> Low-risk women at the onset of labour are attended by their community midwife and have the choice to give birth at home (12.7% of all births in 2019) or in a birth centre or as an outpatient in a hospital (14.6% of all births in 2019) [16]. If problems arise, women are referred to a hospital for prenatal care or during birth.
>
> *Text box*: *The maternity care system in the Netherlands.* [17]

## Methods

### Design

This qualitative study is based on an interpretivist/constructivist paradigm using a framework analysis [18].

### Selection of interventions for vulnerable pregnant women

Included are both interventions specifically developed for vulnerable pregnant women and interventions applied to vulnerable pregnant women that are not specifically developed for this group. During an advisory board meeting (29 January 2019) of the 'together we are strong' projects, attending professionals were asked to provide a list of the interventions they used most often. The 'together we are strong' projects are two projects with the aim to improve the implementation of interventions for vulnerable pregnant women in the northern part of the Netherlands. Both projects share an advisory board that meets at least twice a year. This advisory board involves 25 professionals (midwives, youth nurses, obstetricians, scientists, and an implementation expert) and two lay experts.

The research team selected the interventions that were most frequently mentioned from this list (Box 1).

## Box 1. Description of interventions that are most often offered to vulnerable pregnant women in the North of the Netherlands

| Interventions specifically designed for vulnerable women | |
|---|---|
| ALPHA-NL | The ALPHA-NL is a short questionnaire that all pregnant women fill out at the beginning of their pregnancy and discuss with their midwife. It helps them to talk about their home situation and the circumstances in which the child will grow up. This preventive intervention helps to identify vulnerable pregnant women [19]. |
| Nurse Family Partnership (NFP) (in Dutch: VoorZorg) | This intervention aims to empower and support vulnerable parents. The NFP is an evidence-based child abuse prevention programme. Nurses regularly visit high-risk pregnant women at home during pregnancy until the child's second birthday. There are usually two one-hour visits scheduled per month [20]. |
| Supportive Parenting (in Dutch: Stevig Ouderschap) | This intervention aims to empower and support vulnerable parents. A health visitor makes six home visits during pregnancy and in the first 18 months after childbirth. Supportive Parenting focuses on activating social networks, increasing parenting skills and supporting parent(s)/caregiver(s) to get a good handle on their lives [21]. |
| Mothers Inform Mothers (in Dutch: Moeders Informeren Moeders) | This intervention aims to empower and support vulnerable mothers. Experienced mothers ('buddies') offer voluntary help and support to insecure 'new' mothers during the first two years after childbirth. A paid, highly educated coordinator is responsible for recruiting, selecting, inducting and supervising volunteers as well as for matching volunteers with (expectant) mothers [22]. |
| Meeting Centre for Young Parents (in Dutch: Ontmoetingscentrum voor Jonge Ouders) | This intervention aims to empower and support young parents below 25 years of age. Young parents visit the Meeting Centre for advice, information and to socialize [23]. |
| Interventions applied to vulnerable pregnant women that are not specifically developed for them | |
| Centering Pregnancy | This intervention aims to empower and support existing and prospective parents. The prenatal care group brings women together who are due to give birth around the same time. Over 10 meetings, the group discusses all kinds of health problems with midwives. Women are engaged in their own care by taking their own blood pressure and recording their health data and also have private appointments with their healthcare provider for the monitoring of foetal growth [24]. |
| Trimbos guideline 'Smoking Cessation Support', including V-MIS (Minimal Intervention Strategy for Midwives) | This guideline provides a step-by-step plan for professionals to help pregnant women to stop smoking. It aims to protect the women and their unborn children from tobacco damage and to focus on a healthy lifestyle [25, 26]. |

**Research team and study participants.** The research team consisted of three senior and two junior researchers, all women, from different backgrounds. EdJ-F is a registered midwife, nurse (unregistered), lecturer and healthcare scientist; CW is a psychologist and lecturer; DJ is a nurse (unregistered), lecturer and sociologist; RS is a registered midwife, lecturer and cultural anthropologist; and MD is a registered midwife, lecturer and educational scientist. These various backgrounds represented a diversity of research traditions and methods. Throughout the research team meetings, we reflected on the methodological choices that we made and the perspectives that we chose. This provided us with a deep and broad understanding of our research topic.

Eligible study participants were primary care and hospital-based midwives, gynaecologists, youth and district nurses, regional managers of maternity care assistants, who are using or

have used interventions for vulnerable pregnant women in the North of the Netherlands (in the provinces of Groningen, Friesland and Drenthe). In certain cases, a prior professional relationship between a health professional and a researcher existed before this study started, and the research team reflected on this dynamic. We did not observe any barriers as a result of the dynamic.

**Recruitment.**    We contacted health professionals in our professional network by email, face-to-face and by telephone to ask them to help us recruit participants (snowball sampling). We also asked the coordinator of the Pregnancy and Childbirth North-Netherlands consortium (ZeGNN) to inform all health professionals about this study through its newsletter. Every health professional working with vulnerable pregnant women was eligible to participate. No incentives or compensation for participation in this study was offered. To achieve variation in our sample, final recruitment was targeted at professionals in various categories of adoption of the intervention: innovators, majority, and laggards [10]. Certain groups can be distinguished based on how quickly they adopt an intervention [10]. We deliberately looked for professionals who were 'innovators' and 'early adaptors'. These change agents were the first to pick up new interventions, are well-known leading figures who are respected in the profession and therefore of great importance for dissemination to large groups, the 'majority'. 'Laggards' are afraid of innovation and often need requirements/guidelines from the profession for the final push to adoption. During the iterative process of recruitment and data analysis, we checked whether the voice of the 'majority' and 'laggards' was heard. We also spoke (as a negative case study) with someone who was able to successfully offer an intervention in the Western part of the Netherlands, but was unable to do so in the North of the Netherlands. Other characteristics of healthcare professionals were not taken into account during the recruitment process.

We approached around 80 potential participants. Half of the professionals approached did not respond to the request or indicated that they did not wanted to participate. We aimed to include 40 participants in order to keep the scope of the research topic broad.

**Data collection.**    From February 2019 to April 2020, participants were questioned via in-depth semi-structured interviews about the factors that influence the implementation of interventions for vulnerable pregnant women. We used a topic list (See S1 and S2 Files for the topic list in Dutch and English) for the interviews based mainly on the phases that influence the implementation of interventions as determined by Fleuren et al. [4]. Fleuren et al. describe four critical phases in the implementation process: dissemination, adoption, implementation and continuation. Several factors can influence the transition between these phases positively (facilitators) or negatively (barriers) [4]. The characteristics of the health professional, the intervention, the professional's organizational context and the socio-political environment indirectly determine the intended implementation process [4]. We tested the topic list beforehand for comprehensibility, simplicity and clarity via three pilot interviews. The interviews were conducted by the research team (EdJ-F, CW and RvdS) and by midwifery students, supervised by the research team. They took place at the location of the participant's choice. We conducted interviews until we reached saturation [27, 28].

**Data analysis.**    The audio-recorded interviews were anonymized, transcribed and analysed by the research team. We performed a framework analysis in order to generate practice-orientated findings [29]. First, all of the transcripts were read several times to gain an understanding of the content. An analytical scheme was then developed, inspired by the MIDI (Measurement Instrument for Determinants of Innovations), an instrument designed to identify which determinants influence the actual use of an innovation to be introduced or that had already been introduced [4]. If the research team found determinants that had not been developed a priori, we added them to the analytical scheme.

The research team (EdJ-F, CW, RvdS and MD) jointly processed the results of the analysis and coding them via the computer program ATLAS.ti 8.4. Quotes were translated into English by a certified translator. To assess the credibility of our findings, the final results were discussed at a group session of health professionals and experts by experience on 27 October 2020.

The Consolidated Criteria for Reporting Qualitative Research were used [30].

**Ethical approval and privacy issues.** The study was approved by our institute's Ethics Committee (2019.259). The participants gave written informed consent to the use of information from the interviews. All of the participants were assured of anonymity and confidentiality and could freely withdraw from the study at any time. All data were anonymized. Everyone who came into contact with the research data signed a privacy statement.

## Results

We conducted 36 interviews with 39 participants (33 individual interviews and 3 duo interviews).(See S3 File for the basic characteristics of participating health professionals) The sample consisted of the following participants: community and hospital-based midwives (n = 29), youth and district nurses (n = 3), care coordinators (n = 5), an obstetrician-gynaecologist (ObGyn) and a home counsellor. The age of the participants ranged from 22 to 63 years. The length of experience related to using interventions for vulnerable pregnant women ranged between one and 13 years. The interviews lasted 38 minutes on average (range: 15–85 minutes).

As Fig 1 shows, more than 500 codes were obtained, which were divided into four categories and 26 determinants. Table 1 shows the factors that specifically influence the implementation of interventions for pregnant women in vulnerable situations from the perspective of health professionals.

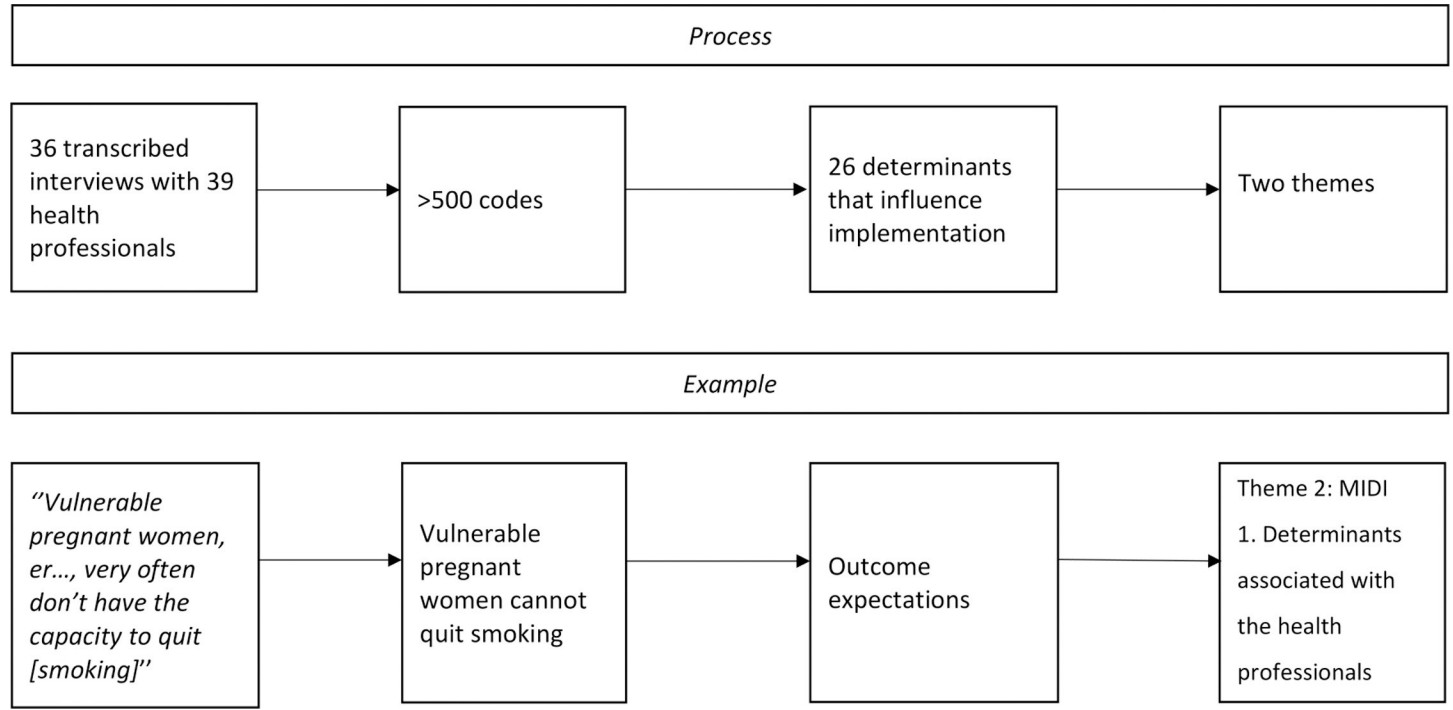

**Fig 1. Process of the analysis: Transcription–Code–Determinant–Theme.**

**Table 1. Factors influencing the implementation of interventions for pregnant women in vulnerable situations, from the perspective of health professionals.**

| Theme 1: Overarching theme | | |
| --- | --- | --- |
| Attitude of the professional | Positive and negative | Attitude is a persistent organization of beliefs, feelings, and behaviours towards socially significant groups, which influences the individuals to respond or behave either positively or negatively to some persons, situations or events [31] |
| **Theme 2: MIDI categories and determinants** | | |
| Categories | Determinants* | Description |
| Determinants associated with the health professional | Reach target group | Being able to reach the intended target group for the intervention |
| | Outcome expectations | Perceived probability and importance of achieving the woman's objectives as intended by the intervention |
| | Woman's satisfaction | Degree to which the health professional expects women to be satisfied with the intervention |
| | Professional obligation | Degree to which the intervention fits in with the tasks for which the health professional feels responsible when doing their work |
| | Woman's cooperation | Degree to which the health professional expects women to cooperate with the intervention |
| | Descriptive norm | Colleagues' observed behaviour; degree to which colleagues use the intervention |
| | Subjective norm | The influence of important others on the use of the intervention |
| | Self-efficacy | Degree to which the health professional believes they are able to implement the activities involved in the intervention |
| | Awareness of content of intervention | Degree to which the health professional has learnt about the content of the intervention |
| Determinants associated with the intervention | Intervention characteristics and feasibility of intervention | Specific features appropriate to the intervention |
| | Familiarity with intervention | Extent to which the user knows about the existence of the intervention |
| | Procedural clarity | Extent to which the innovation is described in clear steps/procedures |
| | Completeness | Degree to which the activities described in the intervention are complete |
| | Complexity of intervention | The degree to which the intervention is complex to implement |
| | Compatibility | Degree to which the intervention is compatible with the values and working method in place |
| | Observability | Visibility of the outcomes for the user |
| | Relevance for women | Degree to which the user believes the intervention is relevant for the pregnant woman |
| Determinants associated with the organizational context of the professional | Collaboration | The collaboration between health professionals and other disciplines |
| | Formal ratification by management | Formal ratification of the intervention by management, for example by including the use of the intervention in policy documents |
| | Replacement when staff leave | Replacement of staff leaving the organization |
| | Staff capacity | Adequate staffing in the department or in the organization where the intervention is being used |
| | Financial resources | Availability of financial resources needed to use the intervention |
| | Time available | Amount of time available to use the intervention |
| | Material resources and facilities | Presence of materials and other resources or facilities necessary for the use of the intervention as intended (e.g. equipment, materials or space) |
| Determinants associated with social-political environment | Influence of the environment and/or culture on target group | Influence of the environment and/or culture on the target group's attitude towards the intervention |
| | Influence of central and/or local government (including legislation and regulations) | Influence of central and/or local government on the implementation of the intervention |

*New determinants (inductive determinants) formulated by the researchers are highlighted in orange.

A number of determinants were aligned with the predefined determinants of the MIDI. In addition, six new determinants were inductively constructed by the researchers (shown

highlighted in orange in Table 1). Eight determinants of the MIDI did not emerge from our analyses. We also found an additional theme in the data which was not directly related to the MIDI, namely professionals' attitude towards and perceptions of the target population. Although this theme is in some way interwoven with the MIDI determinants, it isn't a MIDI category. The reason for describing the attitude of the HCP as a separate theme is because we found this to be such an important issue that has implications not only for the process of implementation but for the overall care process.

First, we present the overarching theme, followed by our second theme, a description of the results related to the four MIDI categories. Quotes have been added to illustrate the professionals' perspectives on and experiences with the factors that influence the implementation of interventions for vulnerable pregnant women. In the interviews, we found many determinants that apply to the implementation process in general. However, due to the large amount of data that we generated, we will only present the determinants that are specifically related to the factors that influence implementation for vulnerable pregnant women.

### Theme 1: Attitude of the health professional

The importance of the attitude of health professionals to the target group was a recurring finding throughout the interviews. Professionals said that it is important to be judgment-free and to be open with the women. A midwife explained that she is a professional who is part of the Centering Pregnancy (CP) group. In order to gain the best outcomes from and the best implementation of this intervention, she needs to be non-judgmental.

> "So, let go of judgement and you'll go a long way. If a woman doesn't feel judged, she can also dare to look a bit deeper into her behaviour or things she does that could possibly be better in terms of health, and then you can achieve more health benefits" (#8, CP).

Health professionals' fear of the stigmatization of vulnerable pregnant women also hinders the implementation of interventions.

> ". . . I need to say how we could do it better" (#26, SCS).

> "It's easy for us to talk with healthcare professionals about 'low socio-economic status', 'vulnerablity'. But as soon as there are clients present, we have serious problems. It feels like a kind of stigmatizing thing, for instance you fit in that box, and because you fit in that box, we need to do all sorts of things with you. We meddle with you" (#17, other).

> "You know, I think we perhaps ask too much [of clients] and maybe intrude on their privacy too much. It makes me think, what if they'd done that to me? Eh? Respect for people, we come very close to them. And that list of risk factors: should we scrap it? Get rid of the judgmental approach" (#19, other).

A negative attitude on the part of the target population was also expressed by professionals, which may be a factor that hinders the implementation of interventions. They said that working with unmotivated and/or vulnerable women can demotivate them, resulting in low participation rates on the part of the women. Some health professionals said that lifestyles in particular are difficult for these women (vulnerable pregnant women) to change, and that many women maintain their poor lifestyles or are at high risk of relapsing into their poor lifestyle habits.

> "This is often a category of people who are absolutely closed to talking about lifestyles" (#36, SCS).

"People who don't want it, you can talk to them as much as you want but it's not going to happen. So that, that's what I don't think is worth investing my time and energy in" (# 29, SCS).

In addition, health professionals used words and phrases in the interviews that could potentially harm, judge or stigmatize women.

"They absolutely stink of cigarette smoke" (#6, CP).

"Identifying pregnant women who are vulnerable: it's a combination of physique, odour, teeth, choice of words" (sniggering) (#3, CP).

## Theme 2: MIDI determinants

**Determinants associated with the health professionals. Being able to reach the target group of the intervention** is of great importance in the implementation of an intervention but, at the same time, it is very difficult according to health professionals.

"Well, you know, the people you want to reach, the people you would really like to have in your group, they don't participate" (#2, CP).

Care avoiders who actually need the most support remain under the radar, according to the professionals, which hinders implementation. It is also difficult to identify the target group properly due to differing characteristics and degrees of vulnerability.

"Well, you know what I think is a very big problem: the actual screening" (#9, NFP/SP).

Health professionals say that reaching vulnerable women is difficult because those women are not always aware of the risks of their unhealthy behavior and therefore need particularly good information for them to understand the need to use an intervention. Raising awareness requires specific skills on the part of health professionals. If a woman is already familiar with an intervention, or the organization that provides the intervention, it is easier for health professionals to encourage women to participate.

"The advantage is that they're already familiar with JGZ and they know they can go there and can get a visit from them" (#14, NFP/SP).

Health professionals mention that they think some interventions do not meet the needs of vulnerable pregnant women, so they decide not to offer them (**outcome expectations**).

"If it really is an addiction, I think it's very difficult to cut down and succeed in such a short time. So many people relapse, No, it's not a nice subject" (#26, SCS).

The degree to which a professional expects women to be satisfied (**women's satisfaction**) with the intervention is another determinant that influences implementation. Professionals mention that they have expectations regarding women's willingness to participate.

(How do people react when you discuss Supportive Parenting?) "It varies: some are very positive, the ones who come and ask for help themselves. Those that we think need help often don't see it like that, they shy away from it" (#13).

Some professionals do not regard counselling, or offering a specific intervention for vulnerable pregnant women, as their **professional obligation**.

"[It] doesn't really feel like my job to go into that in more depth" (#31, SCS).

"So a bit of psychological help, I'm not a psychologist. . . . and addiction care or something like that, I can't give that either, I reckon" (#26, SCS).

Health professionals feel that by implementing additional interventions for vulnerable pregnant women, they are putting regular care in jeopardy; they only feel responsible for giving information and advice and following that up. Other professionals do feel responsible, but incompetent, which also influences implementation.

"Er. . ., yes, whether I really. . ., I think that we do have a responsibility to give guidance, but I don't think I can be responsibile for that right now. Because I'm just not doing it enough" (#33, SCS).

Regarding health professionals' positive or negative expectations about a **woman's cooperation**, they said that women may be concerned about not knowing what to expect of the intervention, or may even be anxious about an intervention. They might express tension because of feared consequences of interventions related to organizations such as 'Safe at Home' (in Dutch: 'Veilig Thuis') or Youth Health Care. This could possibly have effects on whether health professionals offer an intervention.

"So I don't really want to get involved in that again because then, er. . . they'll immediately take, er. . .I'll get an official notification and they'll do something with my child or it'll be taken into care" (#24, SCS).

The **cooperation** of vulnerable women can also be hindered by the possibility of referral to another health professional if they accept an intervention.

"But you notice that some people think they have to go somewhere else. That's a step too far because then they come back and you ask, 'Have you been yet?' Then they say, 'No, not yet.' That's a shame." (#34, SCS).

Additionally, in the eyes of professionals, vulnerable women are more honest but less motivated to cooperate than other women. On the other hand, some vulnerable women do say that they want to join an intervention programme.

"We have a large proportion of parents who say, yes, we'd like to use Supportive Parenting, they're very enthusiastic" (#9, SCS).

**Determinants associated with the intervention.**   According to health professionals, interventions should include specific **characteristics** or need to be feasible to enhance implementation: for example ease of access, voluntary participation and support in practical matters.

". . . you can go along with the mother to the social district team or the school meeting or the municipality . . . looking for a living space" (# 20, other).

"[VoorZorg] really reaches the people in their own surroundings, where the remedial factors can be identified much more readily" (#17, other).

Women may also not participate due to practical reasons such as financial problems, transport or the inability to arrange a babysitter while attending the intervention. For health professionals, the nature of an intervention can be helpful in itself (e.g. the group aspect of CP), as women are more willing to take advice from other women than from a professional. However, the same characteristic of the intervention can also be a hindrance if women are excluded.

"...it seems that other WhatsApp groups sometimes spring up in addition to the original group, resulting in some women being excluded. Because they're, you know, very different from the others" (#6, CP).

**The familiarity** of women with the organization that provides the intervention also encourages vulnerable women to participate, whereas the unfamiliarity of 'new' organizations hinders them from participating.

"The advantage is that they're already familiar with JGZ and they know they can go there and can get a visit from them" (#14, NFP/SP).

Sometimes the activities described in the intervention seem to lack **completeness**. Health professionals say that the intervention lacks options for dealing with resistance and a lack of motivation, or in general how to encourage women to join an intervention programme.

"...but they don't want to quit [smoking] and er... I do find that difficult. Sometimes I think, yes, it would be great if the guidelines included how to deal with the target audience" (#22, SCS).

If health professionals have doubts about the perceived outcomes of an intervention (**observability**), or if they do not receive feedback from other professionals after a referral based on an intervention, they become less motivated to use the intervention as intended.

"Centering Pregnancy was introduced because er... there were so many potential health benefits .... So far, we've run 53 groups, so we've been doing it for quite a while now ... but whether the health outcomes are that much better now... [not sure])..." (#17, other).

"Referral is very easy, but we don't get feedback, so we don't know what happens or what's being discussed there" (#31, SCS).

Health professionals said that interventions can prevent health problems and are therefore relevant for vulnerable woman, which facilitates their implementation.

"I think we should do something about it, as it's very problematic and creates a lot of health risks, or can prevent them if you give a pregant woman proper guidance" (#30, SCS).

**Determinants associated with the organization.** Almost every health professional mentioned the **collaboration** between fellow professionals and with social care professionals as an important factor that influences the implementation of interventions. In the case of vulnerable women, many health professionals take care of these mothers and their babies regarding various problems. The following terms are often mentioned as enhancing the collaboration

between professionals: trust, close contact, clear feedback, key figures as leaders, short lines of communication and a central focal point where professionals can go.

"... in the end, the organizations have the same responsibility but it's the people who are the link. And who know where to find one another" (#19, other).

Interviewees frequently mentioned that poor collaboration between professionals hinders implementation from the start, which is often the case in care provided to vulnerable women.

"Three care workers were involved in caring for that client, they were all operating in isolation, not knowing what the others were doing" (#14, NFP/SP).

The continuity of care professionals for women was also mentioned as an influencing factor.

"The same healthcare providers in one family, you know when there are a lot of changes in the team, it drives you crazy, I think that's a really big problem" (#11, NFP/SP).

A precondition for close collaboration with one another is an overview of the care interventions available for vulnerable pregnant women. Health professionals refer to the fact that there is no social map (a list of names and addresses of agencies involved in the care for vulnerable pregnancies), or only an incomplete one. The social map should include the care content of interventions and the healthcare and social care professionals involved in caring for vulnerable pregnant women.

"I have no idea, er.. .., where to find the municipal debt counselling service. I don't know how to get access to it, it keeps changing all the time, of course" (#4, CP).

The availability of the **financial resources** needed to implement the intervention is one of the essential factors that is most frequently cited. There is often no funding for an intervention.

"Funding is a problem for a lot of midwives. And that's a crying shame" (#10, NFP/SP).

Health professionals mention the lack of **time available** as an important barrier. Supporting vulnerable pregnant women requires more time, due to slower understanding and more extensive problems.

"We did have some vulnerable pregnant women that we spent more time on. When they're slow on the uptake, more problematic, you know the conversation is going to take longer. . . But with that group, I really try to use my relationship of trust more, so I take more time" (#26, SCS).

"Time. . . a very important limiting factor. . ., that's the worst enemy" (#17, other).

**Determinants associated with the socio-political environment.** The **social environment and culture** in which a woman lives can have a positive or negative influence on the implementation of an intervention. Health professionals mention that living in a particular culture can hinder the implementation of interventions and their use as intended: for

example, if women have the impression that they will not fit in with a group such as Center-ing Pregnancy.

> "They're a great community (travellers), but they do feel deprived, just as a result of their culture and experience and the way people deal with them, and the fact that they're not keen to join some civic club. Yes, that is. . . an additional obstacle" (#5, CP).

> "Another one isn't joining in because they heard from those around them, er. . ., 'You don't talk about your problems with someone you just don't know'" (#6, CP).

Many participants describe the impact of the influence of external organizations, such as **local or central government**, i.e. their plans and the funding of interventions. Government support, for example the '1,000-day plan' or SCS guidelines that had been made mandatory for all midwives, is regularly mentioned as beneficial. Health professionals experience an imped-ing influence when an intervention does not have high priority in a municipality or when the funding is not going to be continued.

> "The municipality. . . doesn't really feel any responsibility for that whole group, as I've noticed for the past ten years" (#9, NFP/SP).

> "That subsidy, of course, can give the municipality a push to get it started" (#10, NFP/SP).

> "That whole pot of money, of course. It's a big problem here as well. The municipality is perfectly aware that there's no money but the population that needs it is growing" (#11, NFP/SP).

## Discussion

Our study explored the factors that influence the implementation of interventions for vulnera-ble pregnant women as perceived by health professionals. We found that the attitude of health professionals towards vulnerable pregnant women influences the use or implementation of an intervention. We also noticed that, in general, many barriers and facilitating determinants were mentioned. There was a barrier and a facilitator for almost every determinant. The study revealed 20 MIDI determinants. We added six new determinants to existing MIDI categories. The determinants not presented in the MIDI were related to: being able to reach the target group, intervention characteristics and the feasibility of the intervention, familiarity with the intervention, collaboration between health professionals, the influence of the environment and/or culture on the target group and the influence of central and/or local government.

### Attitude of the health professional

In our study, health professionals exhibited positive and negative attitudes towards vulnerable pregnant women. Some professionals responded to the needs of vulnerable groups and expressed confidence in the women's abilities to learn to cope with their problems and issues. This positive attitude on the part of the professional may enhance implementation and could lead to a positive influence on the use of interventions by other professionals. In line with Jones et al., other studies [11, 31, 32] also show that having a close personal (relation-centred) relationship with a vulnerable group can elicit kindness, concern and compassion, which can be achieved, for instance, by providing a continuity of care.

We also came across stereotyped remarks and words expressing the homogenization of this group of women, which can lead to stigmatization of a group of persons. A negative attitude or

stigma towards vulnerable pregnant women (in this study, mostly native Dutch women living in poverty) can have a considerable influence on women's healthcare-seeking behaviour, engagement of care, adherence to treatment, or may result in women not being offered interventions at all, resulting in a lower quality of prenatal care which is associated with poorer health outcomes (e.g. low birth weight, preterm birth) [33–35].

Women's behaviours and attitudes can frustrate health professionals, leading to negative reactions [36]. In our study, health professionals implicitly and explicitly expressed their frustration by using disrespectful language towards women as a group. A negative attitude of this kind towards certain groups can lead to readily categorizing a person in a particular group with particular characteristics. This categorization may hinder a personal relationship with women [37]. In Briscoe's conceptual model of vulnerability, the role of health professionals can hinder access to care, creating barriers due to a negative professional attitude [38]. In addition to threats and repair mechanisms that influence the degree of vulnerability, the role of the professional can be significant.

Changing the views and attitudes of health professionals is a complex issue. According to Mannava et al. (2015), such attitudes are shaped by interrelated factors: the broader cultural context, working conditions and the workplace environment, provider beliefs and characteristics, the women's attitudes and behaviours and the overall professional-client relationship [36]. In general, attitudes are formed through various processes (social learning, opinions and views of significant others) that are hard to change [39].

## MIDI determinants

The MIDI framework offered a wide range of determinants, providing a good starting point to explore and explain whether the implementation of interventions in existing prevention practices is successful [10]. Many MIDI determinants can be interrelated and can influence one another [9]. For example, reaching the target group as a professional can be hindered by the degree to which the intervention matches the tasks for which the professional feels responsible (professional obligation) or believes that they are able to perform (self-efficacy), or thinks in advance that the woman is not open to an intervention after all or is afraid of being stigmatized (the attitude of the professional). To tackle this, it is important to maintain an overview of the various categories. In addition to the MIDI determinants, we identified six new determinants. These could suggest that they are needed to develop implementation strategies that are more culturally appropriate since we explicitly asked participants about their perceptions of implementing interventions for vulnerable pregnant women.

Health professionals emphasized that a woman's decision whether or not to participate in an intervention has to be voluntary. Important in developing motivation is the need for autonomy, competence and relation-centredness [40]. To make participation voluntary, it is important to be able to have an informed choice. A lack of this can cause feelings of loss of control and loss of autonomy [41]. However, informed choice is not always the case in maternity care in the Netherlands [41].

In the interviews, health professionals mentioned that the social environment and culture that a woman lives in is of great importance and influences successful implementation. Creating greater involvement of the community in the implementation of interventions and ensuring that the values of the culture in which a woman lives are integrated throughout the intervention is important. It creates mutual understanding [11]. Our participants did not mention that greater involvement of the community in the implementation process is necessary. The reason for this may be that health professionals simply do not realize that they can implement interventions in co-creation with the women. Facilitating interaction between

researchers (developers of interventions), health professionals and the women may increase uptake [42] and prevent adverse reactions on the part of vulnerable women to the content of an intervention [8].

Building effective partnerships and collaboration between health providers is pivotal [11]. A lack of collaboration was also an issue in our study (determinant: collaboration) and is defined as a determinant that influences the implementation of culturally appropriate interventions. Collaboration is suboptimal in maternity care in the Netherlands [43]. This could affect the quality of care for all pregnant women [43] and may be of even greater importance for vulnerable women.

A key component in the succesful implementation of interventions is that health professionals have to deliver relation-centred, respectful care [11]. Addressing interpersonal barriers is very important for providing culturally appropriate services. Our participants do not specifically address this issue in those terms but they do say that bonding with the women is very important, while immediately also saying that such bonding is difficult due to a lack of time or the diversity of health professionals who provide care to the women.

## Strengths and limitations

We spoke to just under 40 professionals, a sample size which can be considered quite large in qualitative research [27]. This enabled us to develop a richly textured understanding of highly locally-grounded issues (i.e. saturation) [27]. By using the MIDI in the interviews, we were able to ask about and understand the critical factors that may influence the implementation of interventions for vulnerable pregnant women. From the interviews we collected a very wide range of data. Validation of the MIDI was not a goal, but adding locally grounded information from professionals in the field was our goal. Therefore, we used the MIDI as a template and described each determinant to provide a well-structured analysis of the collected data, which made our research findings more meaningful and generalizable [42].

Having the data coded and analysed by several researchers with different backgrounds and perspectives and discussing the interpretations (investigator triangulation) enhanced the reliability of the findings. Validity was also enhanced using a member-checking approach: the researchers returned the transcripts to the participants to verify the data, and the interpretation of the findings reflected the professionals' experience of implementation.

There are also limitations that needs to be considered when interpreting the results. The interviews were conducted by several researchers, which sometimes caused reduced cohesion between them. We tried to resolve this by holding regular meetings among the research team, at which we discussed the content of the interviews.

Furthermore, our analysis is limited to interventions that are quite successful, as a result we do not provide insight into why some interventions are not finding traction with providers [44]. Moreover, in our study, we only recruited health care providers who used interventions. Future research should assess the perspectives of professionals who do not use interventions designed for vulnerable pregnant women to shed light on the reasons for this non-use. Finally, different themes may emerge when interviewing providers caring for more ethnically diverse populations.

## Recommendations

In order to empower vulnerable women during pregnancy, birth and the postpartum period, a good deal of attention needs to be paid to the attitude of professionals. Exploring and reflecting on one's own attitude, and then changing it if necessary, may be the key to culturally-appropriate implementation of interventions that aim at relation-centred care. Additionally, multiple

determinants influence the implementation of interventions for vulnerable pregnant women at the level of the health professional, the intervention, the organizational and socio-political level, illustrating the complexity of the implementation process. A few steps could be undertaken in research, education and practice to achieve this.

In research, more quantitative data could be collected, for instance by using measures such as the implicit association test, which measures implicit bias [45]. Also, when designing maternal healthcare interventions, the co-creation of interventions can have a positive effect on the attitude of both health professionals and the women for whom the intervention is designed. In this way, interventions can be implemented in a culturally appropriate manner.

In education, it is extremely important to socialize students so that they become health professionals who open up (implicitly and explicitly) [46] to every vulnerable group that they encounter in their daily practice. Practice also needs to pay attention to this issue. Possibly practice and education can join hands to develop a plan together.Throughout the educational curriculum, attention should be paid to challenging stereotypical views and creating positive views of and values towards vulnerable groups [47]. Also, attention needs to be paid in educational programmes and in practice to understanding interventions, the need for them and the complexity of implementing them in a culturally appropriate manner. Using a framework may help to organize and structure the many influencing determinants. A social map that includes all the interventions for vulnerable pregnant women also needs to be developed.

## Conclusions

We highlight the need to challenge stereotypical views and attitudes in order to provide relation-centred care since this is extremely important to provide culturally appropriate care that leads to better outcomes for vulnerable women and their babies. This study showed that many determinants influence the implementation of interventions for vulnerable pregnant women and that many determinants are intertwined, which makes implementation processes very complex. We need to be aware that every woman has a unique social environment in which their values can conflict with those of health professionals, leading to the possible rejection or withdrawal of the care offered.

## Supporting information

**S1 File. Topic list: Interventions for vulnerable pregnant women: Factors influencing implementation according to health professionals: A qualitative study (Dutch).**
(DOCX)

**S2 File. Topic list: Interventions for vulnerable pregnant women: Factors influencing implementation according to health professionals: A qualitative study.**
(DOCX)

**S3 File. Basic characteristics of participating health professionals.**
(DOCX)

## Acknowledgments

The authors would like to thank the health professionals who participated. We are also grateful to the students of the Midwifery Academy Groningen and their supervisors for their support and assistance. We would further like to thank the members of our scientific board and advisory board and our team *Together we are strong*!: Lilian Peters, Stella Weiland, Conny Vreugdenhil and Andrea Drost.

## Author Contributions

**Conceptualization:** Esther I. Feijen-de Jong.

**Data curation:** Esther I. Feijen-de Jong, J. Catja Warmelink, Relinde A. van der Stouwe.

**Formal analysis:** Esther I. Feijen-de Jong, J. Catja Warmelink, Relinde A. van der Stouwe, Maria Dalmaijer.

**Funding acquisition:** Danielle E. M. C. Jansen.

**Methodology:** Esther I. Feijen-de Jong, J. Catja Warmelink, Danielle E. M. C. Jansen.

**Project administration:** Esther I. Feijen-de Jong.

**Resources:** Esther I. Feijen-de Jong.

**Supervision:** Danielle E. M. C. Jansen.

**Writing – original draft:** Esther I. Feijen-de Jong, J. Catja Warmelink, Danielle E. M. C. Jansen.

**Writing – review & editing:** Esther I. Feijen-de Jong, J. Catja Warmelink, Relinde A. van der Stouwe, Maria Dalmaijer, Danielle E. M. C. Jansen.

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
