## [Decision Letter · Decision Letter 0]

22 Oct 2021

PONE-D-21-05965Interventions for vulnerable pregnant women: factors influencing culturally appropriate implementation according to health professionals: a qualitative studyPLOS ONE

Dear Dr. Feijen-de Jong,

Thank you for submitting your manuscript to PLOS ONE. After careful consideration, we feel that it has merit but does not fully meet PLOS ONE’s publication criteria as it currently stands. Therefore, we invite you to submit a revised version of the manuscript that addresses the points raised during the review process. The manuscript has been evaluated by two reviewers, and their comments are available below. The reviewers have raised a number of concerns that need attention. They request additional information on methodological aspects of the study and the interpretation of the results. Could you please revise the manuscript to carefully address the concerns raised? Please submit your revised manuscript by Dec 05 2021 11:59PM. If you will need more time than this to complete your revisions, please reply to this message or contact the journal office at plosone@plos.org. Please include the following items when submitting your revised manuscript:A rebuttal letter that responds to each point raised by the academic editor and reviewer(s). You should upload this letter as a separate file labeled 'Response to Reviewers'.A marked-up copy of your manuscript that highlights changes made to the original version. You should upload this as a separate file labeled 'Revised Manuscript with Track Changes'.An unmarked version of your revised paper without tracked changes. You should upload this as a separate file labeled 'Manuscript'.

We look forward to receiving your revised manuscript.

Kind regards,

Dario Ummarino, Ph.D.

Senior Editor

PLOS ONE

Journal Requirements:

a) Did participants provide their written or verbal informed consent to participate in this study? 

b) If consent was verbal, please explain i) why written consent was not obtained, ii) how you documented participant consent, and iii) whether the ethics committees/IRB approved this consent procedure

3. Please include a copy of the interview guide used in the study, in both the original language and English, as Supporting Information, or include a citation if it has been published previously.

Reviewers' comments:

Reviewer's Responses to Questions

**Comments to the Author**

1. Is the manuscript technically sound, and do the data support the conclusions?

Reviewer #1: Yes

Reviewer #2: Yes

2. Has the statistical analysis been performed appropriately and rigorously? 

Reviewer #1: Yes

Reviewer #2: Yes

3. Have the authors made all data underlying the findings in their manuscript fully available?

Reviewer #1: Yes

Reviewer #2: No

4. Is the manuscript presented in an intelligible fashion and written in standard English?

Reviewer #1: No

Reviewer #2: Yes

5. Review Comments to the Author

Reviewer #1: Review for manuscript PONE-D-21-05965

Thank you for inviting me to participate in the review of this manuscript. The topic is interesting, and the study has been carefully planned and executed. I suggest making some revisions to strengthen the introduction and discussion sections, and re-organizing the results section to make it more focused and readable. Please see specific comments below.

I would also suggest reviewing the manuscript to improve grammar and clarity, possibly with the help of someone with full professional proficiency in English.

Introduction

The introduction could be strengthened by reviewing existing evidence regarding factors that influence implementation. The authors have, undoubtedly, reviewed this literature when devising their research plan and creating their interview guide. I would also appreciate some evidence on the strength of the association between provider factors and the likelihood of implementation. If there is no such evidence, this should also be mentioned.

Study setting

This part is sound, and the authors make compelling arguments for choosing to focus on the Northern part of the Netherlands. It was important to note, as the authors did, that women in this area are mostly of Dutch origin.

I think the authors should also provide some context for routine prenatal care in the Netherlands, as it is different than in many other high-income countries (e.g. midwives led, home birth is much likelier than in other high-income countries etc).

Selection of interventions…

How many professionals were in the meeting in which a list of interventions was created? Was it a focus group, or one on one meetings with these professionals?

Box 1 – description of interventions

This is well-organized, and the interventions are clearly described.

Research team and study participants

Thank you for providing context by describing each researcher’s background. You mention study participants have used/were using interventions for vulnerable pregnant women. Was this a recruitment criterion? If so, please provide reasoning for not including providers who did not use any of the interventions, who could shed light of why some practitioners refrain from implementing such interventions altogether.

Data collection

This section is concise and well-written. As a follow-up to my comment to the previous section, please provide information on inclusion and exclusion criteria, and also on the number of individuals who contacted the research team but then refused or dropped out of the research, in accordance with the COREQ checklist (this can be added to the results section). Were some individuals screened out by the research team, based on their characteristics?

Data analysis

This section is clear and well-written.

Results

Table 1 is well-constructed and clear. Please provide clarification on how “awareness of context of intervention” differs from “familiarity with intervention”.

Theme 1: Attitude of the health professional

Lines 257-267: interviewees 36 and 3 are quoted several times each; was this theme represented broadly among participants?

Theme 2: MIDI determinants

One overarching comment: this section is quite long and could use some editing and re-organization. I suggest summarizing data pertaining to MIDI determinants previously described in a shortened format, and perhaps organize in a table. Lengthier descriptions should be, in my opinion, reserved for the new determinants described by the authors. Another option is to focus solely on the new determinants, provide more detailed descriptions, and mention that previously described determinants have been validated in this study group.

Please provide additional details on the two last determinants: the social environment and culture; and local/central government. Particularly, how do these differ from other determinants (e.g. the examples for government pertain to funding; how does that determinant differ from the funding determinant?).

I’d also be curious to know if interviewees were asked to choose the most salient determinants, or if the researchers noted the most commonly cited determinants, in order to offer some clarity on the relative importance of different determinants.

Discussion

Please provide an explanation to the reason only one determinant (attitude of the health professional) was described separately.

Attitude of the health professional

The term “stigma” is mentioned in the results when describing this determinant, and should also be included in the discussion. I would advise the authors to broaden the discussion of stigma in healthcare, particularly in prenatal care. Also include here the context of the study as interviewees cared mostly for native Dutch population (rather than for immigrants). Consider also including a review of germane literature pertaining to culturally-competent care.

Please provide justification as to why this is an important issue. What do we know of the impact of culturally competent vs. stigmatized care on health outcomes?

Strengths and limitations

Please change the title to strengths and limitation*s*

Using the MIDI may have also “led” interviewees to speak of the experience using the MIDI conceptualization.

Another limitation worth considering is that by selecting the interventions most often used, the authors introduce a bias which should be mentioned in the limitations section (some interventions may not be studied, since they failed to become popular, and factors influencing these interventions will not be studied).

The study has also included health providers caring mostly for native Dutch population. This is both and advantage of the study (indicated that stigma is an important determinant when caring for native, disadvantaged population), and a disadvantage, as different themes may emerge when interviewing providers caring for more ethnically diverse population.

Recommendations

In the last paragraph, you write “above all, attention needs to be paid to the attitude of health professionals…”. This statement requires justification. Is this the determinant found to be most important by the authors? Also, consider collapsing it with the previous paragraph, pertaining to training of health professionals.

Reviewer #2: Overall, the authors address an important research topic: the extent to which personal factors influence the success of health behavior interventions for vulnerable pregnant women. The results suggest that consideration of contextual factors related to attitudes and the degree to which a personal relationship is established influence the effectiveness of the intervention. In addition, organizational barriers were mentioned, as the provision of time and sufficiently trained staff is a prerequisite for successful counselling and collaboration.

Some questions about the research setting remain and need to be addressed to further improve the manuscript.

While the statistical coding and analysis was done carefully and systematically, participants were recruited through a snowball system. This sampling raises questions about the extent to which respondents are representative of all professional groups working in the field under study (only one obstetrician, and is a mother (lay person?) from the 'mother Inform mothers' concept in the sample?) Although many results may be generalizable, a sample stratified by occupational groups and length of time working in the field might reveal more specific results.

For example, ‘bonding’ with clients is described as helpful in improving health behavior. However, it does fit less to the role of the doctor (obstetrician) than to the ‘mother to mother’- concept, which seems more promising to reach pregnant women through the peer effect. Therefore, the experiences of health workers (such as midwifes, doctors and nurses) in building relationshops with women, may differ from the findings of more peer-orientated staff (including ‘mothers inform mothers’). The difference should be presented and discussed as it may require adapted improvements strategies. In addition, the used intervention tools vary from specific interventions for vulnerable pregnant women to general, e.g. specific smoking cessation. Do these differences not affect the respective goals and outcomes, which would then have to be evaluated diffently?

Another issue is the length of employment in this area of work, as both experience an the potential for frustration increase over time. This should also be taken into account when presenting the results. Is there a trend to be observed here? This information could be supportive to implement the research results into improved practice.

Furthermore, the semi-structured questionnaire and a demographic table with information about gender, professional groups, and length of work experience should be added to display a clear picture of participants.

6. PLOS authors have the option to publish the peer review history of their article (what does this mean?). If published, this will include your full peer review and any attached files.

Reviewer #1: No

Reviewer #2: No

---

## [Author Response · Author response to Decision Letter 0]

13 Dec 2021

We added our response to the reviewers in a file which we uploaded.

---

## [Decision Letter · Decision Letter 1]

2 Mar 2022

PONE-D-21-05965R1Interventions for vulnerable pregnant women: factors influencing culturally appropriate implementation according to health professionals: a qualitative studyPLOS ONE

Dear Dr. Feijen-de Jong,

Thank you for submitting your manuscript to PLOS ONE. After careful consideration, we feel that it has merit but does not fully meet PLOS ONE’s publication criteria as it currently stands. Therefore, we invite you to submit a revised version of the manuscript that addresses the points raised during the review process.

 The reviewers have reassessed the manuscript and find it improved. However, they believe that further revisions can be made to improve the quality of reporting and presentation of the Discussion. Their comments are appended below. 

Could you please revise the manuscript to carefully address the concerns raised?

We look forward to receiving your revised manuscript.

Kind regards,

Marianne Clemence

Staff Editor

PLOS ONE

Journal Requirements:

Reviewers' comments:

Reviewer's Responses to Questions

**Comments to the Author**

1. If the authors have adequately addressed your comments raised in a previous round of review and you feel that this manuscript is now acceptable for publication, you may indicate that here to bypass the “Comments to the Author” section, enter your conflict of interest statement in the “Confidential to Editor” section, and submit your "Accept" recommendation.

Reviewer #1: (No Response)

Reviewer #2: All comments have been addressed

2. Is the manuscript technically sound, and do the data support the conclusions?

Reviewer #1: Yes

Reviewer #2: Yes

3. Has the statistical analysis been performed appropriately and rigorously? 

Reviewer #1: Yes

Reviewer #2: Yes

4. Have the authors made all data underlying the findings in their manuscript fully available?

Reviewer #1: Yes

Reviewer #2: Yes

5. Is the manuscript presented in an intelligible fashion and written in standard English?

Reviewer #1: Yes

Reviewer #2: Yes

6. Review Comments to the Author

Reviewer #1: Review for manuscript PONE-D-21-05965-R1

Thank you for sending the revised manuscript. I believe the revisions have greatly strengthened the manuscript and have few minor comments to make.

Methods

Study Setting

Thank you for adding a text box contextualizing the maternity care system in the Netherlands. I believe it significantly adds to the understanding of international readers. Please add a reference to the text box in the text of this section.

Selection of interventions

Thank you for clarifying the setting in which interventions were selected. I would add to the manuscript text all details you provided in your response to my comment (i.e., not only the number of participants, but also that this was a part of an advisory board meeting for a project) and also an explanation of the nature of the Together we are strong project. This sounds like a very appropriate setting in which to collect the list of interventions – I just want all details to be clear for readers (e.g. if they want to replicate a similar intervention selection process in a different country).

Research team and study participants

Thank you for adding information both to the methods section and to the discussion on study participants being limited to those who use the interventions. I would even be more explicit in the discussion (e.g. “our study only recruited participants who used interventions, the perspective of practitioners who do not use any intervention should be assessed etc”), as some people will not read the methods section carefully.

Data collection

Thank you for the modifications. Please add the absolute number of practitioners approached (was it 80? 81? 79?). Please also add information on how the categories of adoption of intervention were assessed – this is super interesting and a unique strength of your study. Please also mention in the text no individuals were screened out by the study team.

Theme 2: MIDI determinants

a. The explanation for keeping all MIDI determinants in is sound and reasonable. Please add a short version of this explanation to the manuscript. Please clearly separate between existing MIDI determinants and new determinants described by the authors.

Discussion

Strengths and limitations

c. In my comment, I referred to interventions that failed to become popular, and were therefore not mentioned. This is a different limitations than the one related to the study participants. I would also add this to the limitations (your analysis is limited to interventions that were quite successful to begin with, and does not shed light of why some interventions gain no traction among providers).

* Thank you for including your interview guide and characteristics of study participants as supporting information. I suggest to include the characteristics of study participants in the manuscript (if table number limits allow it), and in any case, to include a reference to these two important sources in the manuscript text.

Reviewer #2: The authors have addressed my suggestions. Nethertheless, one point remains about the background of the participants.

In the resubmission was clarified, that only professional health care worker were included. What about women from the “mothers inform mothers” approach? Is this intervention conducted by lay persons or professionals, too? This should be specified.

7. PLOS authors have the option to publish the peer review history of their article (what does this mean?). If published, this will include your full peer review and any attached files.

Reviewer #1: No

Reviewer #2: No

---

## [Author Response · Author response to Decision Letter 1]

28 Mar 2022

within the response to the reviewers document we responded to the comments of the reviewers. We are grateful for the comments we have received. Because of that we were able to strengthen the manuscript.

---

## [Decision Letter · Decision Letter 2]

26 Apr 2022

PONE-D-21-05965R2Interventions for vulnerable pregnant women: factors influencing culturally appropriate implementation according to health professionals: a qualitative studyPLOS ONE

Dear Dr. Feijen-de Jong,

Thank you for submitting your revised manuscript to PLOS ONE. After careful consideration, we feel that it has merit but does not fully meet PLOS ONE’s publication criteria as it currently stands. Therefore, we invite you to submit a revised version of the manuscript that addresses the points raised during the review process. Your manuscript has been reviewed by both original reviewers, whose comments are appended below. Reviewer 1 is satisfied by the revisions and recommended acceptance. However, Reviewer 2 raises a minor concern that in their review was not fully addressed by the prior revision. Please ensure that this point is addressed in full before resubmission.

We look forward to receiving your revised manuscript.

Kind regards,

Emily Chenette

Editor in Chief

PLOS ONE

Journal Requirements:

Reviewers' comments:

Reviewer's Responses to Questions

**Comments to the Author**

1. If the authors have adequately addressed your comments raised in a previous round of review and you feel that this manuscript is now acceptable for publication, you may indicate that here to bypass the “Comments to the Author” section, enter your conflict of interest statement in the “Confidential to Editor” section, and submit your "Accept" recommendation.

Reviewer #1: All comments have been addressed

Reviewer #2: (No Response)

2. Is the manuscript technically sound, and do the data support the conclusions?

Reviewer #1: Yes

Reviewer #2: Yes

3. Has the statistical analysis been performed appropriately and rigorously? 

Reviewer #1: N/A

Reviewer #2: Yes

4. Have the authors made all data underlying the findings in their manuscript fully available?

Reviewer #1: Yes

Reviewer #2: Yes

5. Is the manuscript presented in an intelligible fashion and written in standard English?

Reviewer #1: Yes

Reviewer #2: Yes

6. Review Comments to the Author

Reviewer #1: (No Response)

Reviewer #2: The authors did not address my comment on the first resubmission. The manuscript states that only professional health care workers were included. The concept of "mothers for mothers" reads like a lay approach, so please clarify, especially for the international readership, whether or to what extent this group is trained and/or paid for their work in addition to their personal experience as a mother.

7. PLOS authors have the option to publish the peer review history of their article (what does this mean?). If published, this will include your full peer review and any attached files.

Reviewer #1: No

Reviewer #2: No

---

## [Author Response · Author response to Decision Letter 2]

13 May 2022

We uploaded our respons to the reviewers.

---

## [Decision Letter · Decision Letter 3]

18 Jul 2022

Interventions for vulnerable pregnant women: factors influencing culturally appropriate implementation according to health professionals: a qualitative study

PONE-D-21-05965R3

Dear Dr. Feijen-de Jong,

We’re pleased to inform you that your manuscript has been judged scientifically suitable for publication and will be formally accepted for publication once it meets all outstanding technical requirements.

Kind regards,

Alejandra Clark

Division Editor

PLOS ONE

Additional Editor Comments (optional):

Reviewers' comments:

Reviewer's Responses to Questions

**Comments to the Author**

1. If the authors have adequately addressed your comments raised in a previous round of review and you feel that this manuscript is now acceptable for publication, you may indicate that here to bypass the “Comments to the Author” section, enter your conflict of interest statement in the “Confidential to Editor” section, and submit your "Accept" recommendation.

Reviewer #2: All comments have been addressed

2. Is the manuscript technically sound, and do the data support the conclusions?

Reviewer #2: Yes

3. Has the statistical analysis been performed appropriately and rigorously? 

Reviewer #2: Yes

4. Have the authors made all data underlying the findings in their manuscript fully available?

Reviewer #2: Yes

5. Is the manuscript presented in an intelligible fashion and written in standard English?

Reviewer #2: Yes

6. Review Comments to the Author

Reviewer #2: The authors have adequately answered my question about the professional backgrounds of the participants of the 'mothers for mothers' approach. Thank you!

7. PLOS authors have the option to publish the peer review history of their article (what does this mean?). If published, this will include your full peer review and any attached files.

Reviewer #2: **Yes: **Dr. Martina Schmiedhofer MPH, Senior Researcher at Jacobs University Bremen and Charité-Universitätsmedizin Berlin

---

## [Editor Report · Acceptance letter]

21 Jul 2022

PONE-D-21-05965R3 

Interventions for vulnerable pregnant women: factors influencing culturally appropriate implementation according to health professionals: a qualitative study 

Dear Dr. Feijen-de Jong:

I'm pleased to inform you that your manuscript has been deemed suitable for publication in PLOS ONE. Congratulations! Your manuscript is now with our production department. 

Kind regards, 

on behalf of

Dr. Alejandra Clark 

Staff Editor

PLOS ONE